# Bacillamide F, Extracted from Marine *Bacillus atrophaeus* C89, Preliminary Effects on Leukemia Cell Lines

**DOI:** 10.3390/biology11121712

**Published:** 2022-11-25

**Authors:** Shengnan Zhang, Giorgia Croppi, Heng Hu, Yingxin Li, Chunmiao Zhu, Fang Wu, Fengli Zhang, Zhiyong Li

**Affiliations:** 1Marine Biotechnology Laboratory, State Key Laboratory of Microbial Metabolism, School of Life Sciences & Biotechnology, Shanghai Jiao Tong University, Shanghai 200240, China; 2Key Laboratory of Systems Biomedicine (Ministry of Education), Shanghai Center for Systems Biomedicine, Shanghai Jiao Tong University, Shanghai 200240, China

**Keywords:** bacillamide, *Bacillus atrophaeus*, leukemia, antiproliferation effect

## Abstract

**Simple Summary:**

Bacillamides are important bioactive natural compounds. This paper depicts for the first time a study of a novel thiazole alkaloid compound bacillamide F that was isolated from the marine *Bacillus atrophaeus* C89. Bacillamide F inhibits the leukemia cell lines HL60 and Jurkat. The work provides a new indication of the pharmacological activity of bacillamides.

**Abstract:**

Developing new treatments for leukemia is essential since current therapies often suffer from drug resistance and toxicity. Bacillamides are very promising, naturally occurring compounds with various bioactivities. In the present study, we investigated the use of bacillamide analogues, a new thiazole alkaloid bacillamide F that was isolated from marine *Bacillus atrophaeus* C89 associated with sponge *Dysidea avara*. The structure of the new compound bacillamide F with indolyl–thiazolyl–pyrrolidine ring was determined by high resolution mass spectrometry, secondary mass spectrometry, and nuclear magnetic resonance analyses. Intriguingly, bacillamide F is able to inhibit the proliferation of an acute myeloid leukemia cell line HL60 (IC50 (24 h) 21.82 µM), and an acute T-cell leukemia Jurkat (IC50 (24 h) 46.90 µM), rather than inhibit the proliferation of the acute histiocytic lymphoma U-937 cell line, human fetal lung fibroblast MRC-5 cell line, and some solid tumor cell lines (IC50 (24 h) > 100 µM). The study provides a new indication of the pharmacological activity of natural product bacillamides.

## 1. Introduction

Leukemia is a disease that develops from bone marrow, and can be classified into acute leukemia and chronic leukemia [1]. In regard to the type of cells involved in the disease, leukemia is further divided into acute myelogenous leukemia (AML) [2], chronic myeloid leukemia (CML) [3], acute lymphoblastic leukemia (ALL) [4], and chronic lymphoblastic leukemia (CLL) [5]. Moreover, each class of leukemia contains several subtypes [6]. For instance, AML can be subdivided into more than ten different subtypes according to the defining genetic abnormalities in the new WHO (World Health Organization) classification scheme. Due to the many different types of leukemia and fast progress of resistance of certain subtypes for current therapy, new treatments are becoming necessary [7,8].

Bacillamides, which are non-ribosomal peptides thiazole alkaloids, can be categorized as bacillamide A, B, C, D and E (Figure 1a) [9,10,11,12,13,14]. Bacillamide A was isolated from the marine bacterium *Bacillus* sp. SY-1 [9]. Bacillamide B was isolated from *Bacillus endophyticus* [10] and *Microbispora aerate* IMBAS-11A [11]. Bacillamide C was isolated from *Bacillus endophyticus* [10] and *Bacillus atrophaeus* C89 associated with marine sponge *Dysidea avara* [12]. Bacillamide D was isolated from *Thermoactinomyces* sp. strain TM-64 [13]. Bacillamide E was synthetized in vitro by Bloudoff et al. [14]. Bacillamides and their derivatives show significant algicidal activities against a wide range of dinoflagellates, raphidophytes, and particular species of cyanobacteria [9,15], and bacillamide analogues show cytotoxic and anti-inflammatory activities [16]. Moreover, thiazole derivatives exhibit good anticancer activity with influence on cell cycle, DNA fragmentation and mitochondrial depolarization [17].

Precursor-directed biosynthesis (PDB) was an effective method used to generate novel NRPS products analogues [18]. In our previous study, substrate selection of adenylation domains for non-ribosomal peptide synthetase (NRPS) in bacillamide C biosynthesis by *B. atrophaeus* C89 was investigated, and the results proved the A1 and A2 domains can catalyze a variety of substrate amino acids, including proline [19]. Here, we isolated a novel thiazole alkaloid bacillamide F and found that the production of bacillamide F was increased through the addition of precursor L-proline into a culture medium of marine *B. atrophaeus* C89. We elucidated the structure and inhibition activity of bacillamide F in this research. To our knowledge, this is the first report of bacillamide analogue bearing a 2-(thiazole-2-yl-) pyrrolidine, and a biosynthetic pathway of bacillamide F was predicted. In this study, the antiproliferation effect on leukemia of bacillamide F was investigated, and the results showed that bacillamide F inhibited the proliferation of leukemia cell lines Jurkat and HL60.

## 2. Materials and Methods

### 2.1. Biosynthesis and Extraction of Bacillamide F

*B. atrophaeus* C89 (CCTCC AB 2016282, Genome Genebank No, AJRJ00000000.1) was isolated from the sponge *D. avara* in the South China Sea [12]. *B. atrophaeus* C89 was incubated at 28 °C in a solid medium containing 5 g of beef extract, 10 g of peptone, and 20 g of agar in every 1000 mL of artificial seawater (ASW) with pH 7.0–7.2. After 24 h cultivation, the strain was cultured in a modified medium (consisting of 3.64 g/L corn starch, 6.29 g/L CaCO_3_, 4.00 g/L soy peptone, and 6.00 g/L peptone with ASW) at 28 °C for 96 h; and 0.10 g/L cysteine, 0.02 g/L tryptophan, and 0.02 g/L proline were added to the culture medium when the strain was cultured at 40 h.

After 96 h cultivation, the medium (18 L) was extracted three times with ethyl acetate (EtOAc). Evaporation of EtOAc extracts gave the yellow oil (3.40 g) which was subjected to Sephadex LH-20 columns with methanol (MeOH). Consequently, 2 fractions (Fr. 1–2) were obtained according to Thin Layer Chromatography (TLC) monitoring. Fr. 2 (2.40 g) was further purified on Medium Pressure Preparative Liquid Chromatography (MPLC). As a result, 8 fractions (Fr. 3–10), according to reversed-phase high-performance liquid chromatography (HPLC) monitoring (1 mL/min, detector UV λ_max_ 220 nm, acetonitrile/H_2_O 20: 80–45: 55, 15 min), were eluted with increasing proportions, respectively, of MeOH (H_2_O/MeOH (*v*/*v*, 90:10, 70:30, 50:50, 30:70, 10:90, 0:100)). Finally, Fr. 8 (136 mg) was further purified on semi-preparative HPLC (1 mL/min, detector UV λmax 220 nm, acetonitrile/H_2_O 30: 70) to deliver bacillamide F (34.80 mg).

### 2.2. General Experimental Procedures

Column chromatography (CC): Sephadex LH-20 (Amersham Biosciences, Amersham, UK). MPLC: Cheetah Fs-9200t (Bonna-Agela, Tianjin, China). TLC: precoated silica gel plates (Qingdao Marine Chemical Company, Qingdao, China). Semi-preparative HPLC: Agilent 1200 series liquid chromatograph Durashell C18-AM column (4.60 × 250 mm; i.d. 5 µm). Reversed-phase HPLC: Agilent 1200 series liquid chromatograph ZORBAX Eclipse XDB-C18 column (4.60 × 150 mm; i.d. 5 µm). Mass spectra: ACQUITYTM UPLC & Q-TOF MS Premier instrument equipped with an electrospray ionization (ESI) probe operating in positive-ion mode with direct infusion. ^1^H and ^13^C- Nuclear Magnetic Resonance (NMR) spectra: Avance III 600 MHz spectrometer; chemical shifts δ in ppm, with residual methanol-d4 (δ_H_ 3.31, δ_C_ 49.15) as internal standard, coupling constant J in Hz. ^1^H- and ^13^C-NMR assignments were supported by ^1^H, ^1^HCOSY, Heteronuclear multiple quantum coherence (HMQC), heteronuclear multiple bond correlation (HMBC) experiments. All chemicals used in the study were analytical grade and HPLC grade.

### 2.3. Cell Culture

The acute T-cell leukemia cell line (Jurkat cells) was a generous gift from Prof. H. Gehring (Department of Biochemistry, University of Zürich, Switzerland). The histiocytic lymphoma cell line (U-937 cells) and acute promyelocytic leukemia cell line (HL60 cells) were generous gifts from Dr. H. Le (Shanghai Center for Systems Biomedicine, Shanghai Jiao Tong University). The cells were cultured in RPMI-1640 medium (Invitrogen, Waltham, MA, USA) supplemented with 10% of fetal bovine serum (Lonza, Basel, Switzerland) and 1% of penicillin-streptomycin antibiotics. The cells were grown at 37 °C in a humidifier incubator under an atmosphere of 5% CO_2_ and used for assay when they reached 80% of confluence in the dish.

### 2.4. MTS Assay

The viability of the cells was measured by MTS, a colorimetric assay which involves the biological reduction of the MTS reagent tetrazolium (3-(4,5-dimethylthiazol-2-yl)-5-(3-carboxymethoxy-phenyl)-2-(4-sulfophenyl)-2H-tetrazo-lium). Cells were seeded in a 96-well plate at 5000 cells/well density. Bacillamide F was added at different concentrations (6.25–100 µM) and the cells were treated for 24, 48 and 72 h. Absorbance was recorded at 490 nm in a plate reader (BioTek & Synergy2, Singapore). The number of cells in experiment is the means from three wells cells (biological replicates). All the experiments were repeated at least twice.

## 3. Results

### 3.1. Structural Identification of New Compound

The EtOAc extract of the bacterium was subjected to repeated Sephadex LH-20; the pure compound 1 (Figure 1b) was obtained by MPLC and semipreparative HPLC.

Compound 1 was obtained as yellow oil. It showed a molecular ion peak at *m*/*z* 383.1549 [M + H]^+^ (calcd. for C_20_H_22_N_4_O_2_S, 383.1549) in the positive HR-ESI-MS spectrum, indicating twelve degrees of unsaturation. In the ^1^H-NMR spectrum (Appendix A), five signals for H-atoms between δ_H_ 6.97 and 7.58 indicated the presence of one indole ring. Furthermore, one aromatic H-atom at δ_H_ 8.03 (s, 1 H), one CH group at δ_H_ 5.32 (dd, J = 2.80, 7.60, 1 H), five CH_2_ groups at δ_H_ 3.05 (t, J = 7.20, 2 H), 3.68 (t, J = 7.20, 2 H), 2.23 (m, 2H), 2.03 (m, 2H) and 3.71~3.58 (m, 2H), and one Me group at 2.09 (s, 3 H) were observed. The ^13^C-NMR and Distortionless Enhancement by Polarization Transfer (DEPT) spectra exhibited 20 C-atom signals (1 × CH_3_, 5 × CH_2_, 7 × CH, 5 × C, and 2 × C=O), whose chemical-shift values and multiplicities confirmed the presence of an indole ring (δ_C_ 123.57, 113.04, 119.37, 119.65, 122.38, 112.24, 138.20, 128.77). The ^1^H-^1^H COSY spectrum of H-9 (δ_H_ 3.05)/H-10 (δ_H_ 3.68), H-15 (δ_H_ 5.32)/H-16 (δ_H_ 2.23)/H-17 (δ_H_ 2.03)/H-18 (δ_H_ 3.71~3.58) revealed that two spin systems were present in compound 1 (Figure 2, Appendix A).

HMBC correlations from H-9 and H-10 to C-2 suggested that C-9 was connected to the indole ring. The other significant HMBC cross-peaks from H-13 to C-11, C-12, and C-14, from H-18 to C-14 and C-15, and from Me-20 to C-19 confirmed the connectivity of the pyrrolidine and thiazole rings as shown in Figure 2. Considering the molecular formula, the presence of a disubstituted thiazole ring was suspected and C-18 was connected to C-15 through a N-atom, while an acetyl group was anchored at the N-atom, further verified by HMBC correlation from H-15 to C-19. The compound 1 was identified as bacillamide F by analysis of its NMR spectral data as well as by comparison with the reported data [11]. Thus, the planar structure of compound 1 was established.

### 3.2. Plausible Biosynthesis Pathway of Bacillamide F

According to genome information, bacillamide F structure, and predicted biosynthetic pathway of bacillamide C and intermediate AlaCysthiazole [19,20], biosynthesis of bacillamide F has been predicted (Figure 3). NRPS domains, Oxidase domain, and AADC (aromatic L-amino acid decarboxylase) domain have been involved in the synthesis of bacillamide F. The enzyme-bound amino acids are combined through a nucleophilic attack of either the cysteine amino group or the heteroatom of the side chain onto the carbonyl C of proline, and a proton is abstracted through base catalysis, thereby enabling the attack of either the cysteine side chain or the amino group onto the amide bond carbonyl. Intermediate hydroxylated thiazolidine has been synthesized, and then hydrated and dehydrogenated to produce thiazole-containing product ProCys_thiazole_. Bacillamide F precursor has been synthesized by the C domain catalyzation of peptide bond formation between the ProCys_thiazole_ and L-tryptamine decarboxylated by the AADC enzyme; subsequently, bacillamide F is synthesized after acetyl modifications.

In our previous study, a possible biosynthesis pathway for bacillamide C suggested that bacillamide C could be derived from alanine, cysteine, and tryptophan [19,21]. Consistent with this hypothesis, and followed by the optimization of the culture medium, experimental results showed that the independent addition of alanine, cysteine, and tryptophan had a positive effect on production of bacillamide C [22]. Based on the predicted biosynthetic pathway, it is possible that the pyrrolidine in compound 1 derived from L-proline (Figure 3). Therefore, the prediction of biosynthesis pathway for bacillamide F benefits the production of bacillamide F through optimized medium. In our other experiment, the increase of biosynthesis of bacillamide F was detected and quantified when the *B. atrophaeus* fed on L-proline.

### 3.3. Toxicity of Bacillamide F on Leukemia Cells

To determine the potential effect of the new compound on cancer cells, we performed a toxicity assay (MTS) on three different cell lines of leukemia. Bacillamide F inhibited the growth of Jurkat in a dose-dependent manner (Figure 4), showing cell survival less than 20% at 50 µM after 72 h of treatment. HL-60, acute promyelocytic leukemia, the subtype of AML, exhibited growth suppression in a dose- and time-dependent manner with a survival rate less than 30% at 12.50 µM after 72 h, while bacillamide F did not show any suppression on U937 as well as fibroblast MRC-5 or other solid tumor cells, such as pancreatic cancer cell lines PATU8988 and SW1990. Cytotoxicity of the inhibitor on Jurkat and HL60 was found to be decreased time-dependently with an IC50 of 33.09 µM and 9.44 µM after the treatment of 72 h, an IC50 of 37.16 µM and 14.13 µM after 48 h, and an IC50 of 46.90 µM and 21.82 µM after 24 h, respectively (Appendix A). Intriguingly, bacillamide F showed a much less inhibitory effect on the proliferation of Jurkat than on HL60, e.g., at the concentration of 25 µM (Figure 4), an effect that we cannot currently explain.

Previous results have revealed that synthetic analogues of bacillamide type compounds with structural modifications on the amino or aminophenyl group of position 2 of the thiazole ring could suppress the proliferations of both solid tumor and leukemia cell lines [16]. In the present study, we show that a newly identified natural product bacillamide F, which has a substitution of a methyl-amino group on the same position of the thiazole ring, can inhibit the proliferation of ALL and acute promyelocytic leukemia (APL) cell lines Jurkat and HL60 without affecting the proliferation of the solid tumor cell line (Appendix A). The attractive indication of bacillamide F provides a pharmacological activity for this type of natural compound and offers a new drug lead for the treatment of ALL and APL leukemias.

## 4. Discussion

Bacillamides and analogues are natural products with various bioactivities, including algicidal activity [9,15,23], cytotoxic activity against three cancer cell lines (HCT-116, MDA-MD-231, and Jurkat cell lines), and anti-inflammatory activity [16]. Indole alkaloid derivatives, as the building blocks of bacillamides, have exhibited antibacterial and antifungal activity [24]. Thiazole amides, the component of bacillamides, have exhibited potential algicidal activity against harmful cyanobacterial algae [23]. Bacillamides, as a building block, have been found in cyclic peptides antibiotic zelkovamycin and argyrin A and B [25,26]. In this study, bacillamide F controlled the growth of Jurkat ALL and AML HL60 cell lines in a dose-dependent manner, rather than the acute histiocytic lymphoma U937 cell line (Figure 4 and Appendix A), which is also considered to be a different subtype of AML [27]. Bacillamide F has a pyrrole structure (Figure 1b). Pyrrole is a simple heterocyclic and important building block for biologically active compounds [28]. The anticancer activity of bacillamide F has broadened the active range of the bacillamides family. Although bacillamide F shows a promising growth-inhibitory effect in two types of leukemia cell lines, further work needs to be performed in vitro and in vivo models to investigate its function and molecular mechanism in these cells.

The biosynthetic pathway of bacillamide has been predicted in our previous study [19,20,21]. The biosynthesis of bacillamide C has been hypothesized that decarboxylation from L-tryptophan to tryptamine could be performed before amidation by the downstream aromatic AADC to nrps gene cluster [21]. We have predicted that Oxidase encoded by oxidase gene located upstream of nrps gene cluster could catalyze AlaCys_thiazoline_ to AlaCys_thiazole_ [19]. The dehydrogenation activity of Oxidase has been further confirmed in the peptide intermediate while it is covalently attached to the NRPS [29]. In this study, addition of L-proline into a culture medium of *B. atrophaeus* C89 increased the production of bacillamide F (Figure 2). These results implied that broad substrate selectivity of NRPS A domain probably contributes to the biosynthesis of bacillamide analogues. In biosynthesis of bacillamide F as predicted, alanine, cysteine, tryptophan, and proline are precursors of bacillamide F (Figure 3). In future work, we will investigate how the concentration and proportion of various amino acid precursors affect the production of bacillamides and analogues.

## 5. Conclusions

In this study, a new member of the bacillamide family, bacillamide F, was obtained by a series of separation and purification methods. NMR data showed that the compound contained a novel thiazole alkaloid structure. The study also revealed that bacillamide F inhibited the growth of leukemia cell lines HL60 and Jurkat.

## Figures and Tables

**Figure 1 biology-11-01712-f001:**
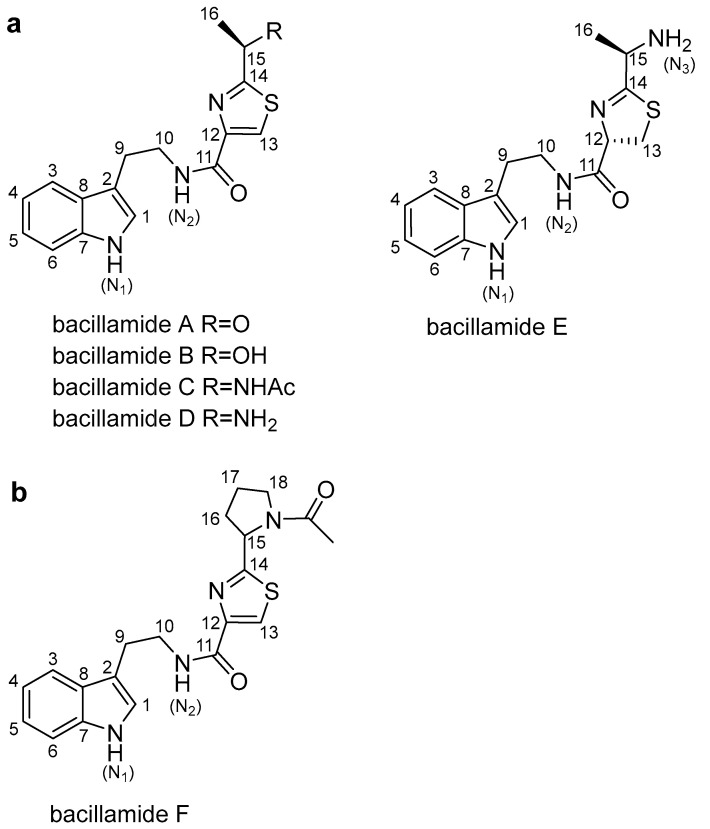
The bacillamide family structures of compounds: (**a**) This represents the known bacillamide compounds (bacillamide A–E); (**b**) This represents the new compound, bacillamide F.

**Figure 2 biology-11-01712-f002:**
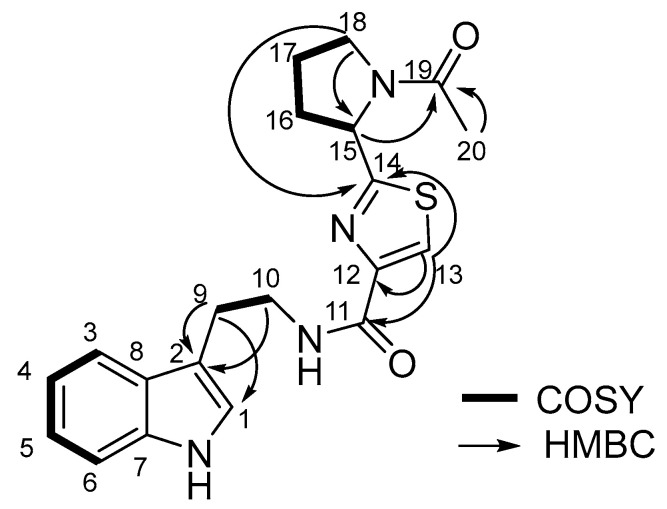
The key 2D NMR correlations of bacillamide F.

**Figure 3 biology-11-01712-f003:**
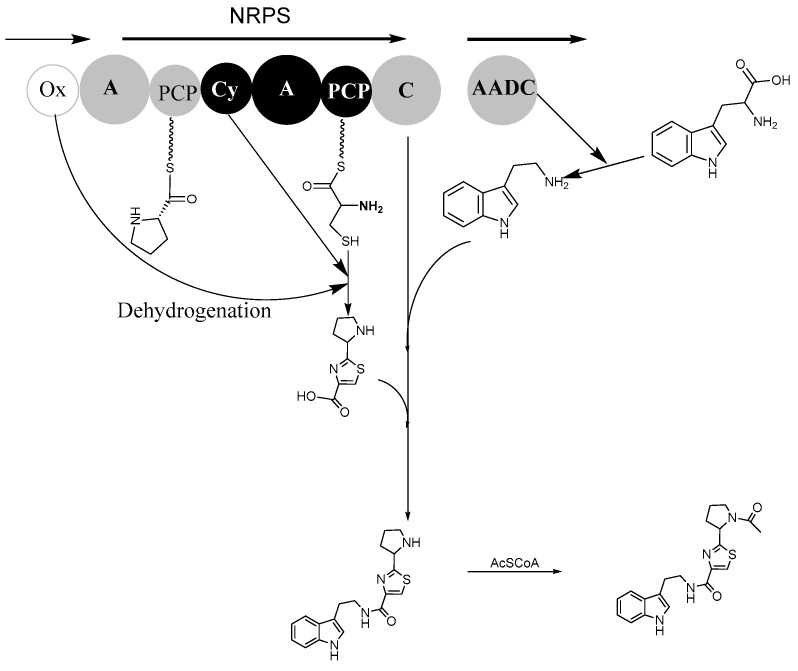
The plausible biosynthesis pathway of bacillamide F. The biosynthetic process of NRPS (EIM09914.1, bacitracin synthetase) domains, Ox (oxidase domain, EIM09913.1), and AADC (aromatic L-amino acid decarboxylase) domain were involved in the synthesis of bacillamide F. Each circle represents an NRPS enzymatic domain, from left to right: A, adenylation domain, Cy, cyclization domain. C, condensation domain, and PCP, peptidyl carrier protein domain.

**Figure 4 biology-11-01712-f004:**
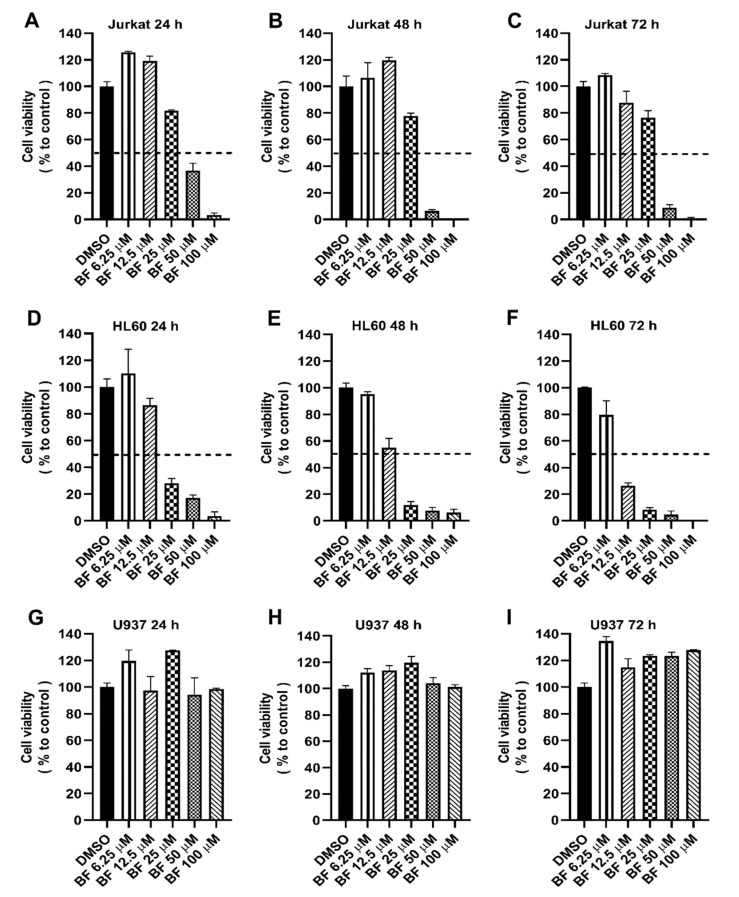
MTS assay on three different leukemia cell lines: Jurkat (**A**–**C**), HL60 (**D**–**F**), and U937 (**G**–**I**). Cell availability has been tested for 24, 48 and 72 h by incubating the cells in a 96-well plate with different concentrations of BF (Bacillamide F) (6.25–100 µM). The percentage of control samples (DMSO, 100%) is shown as means ± SDs (*n* = 3, biological replicates). The experiments were repeated at least twice independently with similar results, and the representative data from one experiment are shown (*n* = 3, biological replicates).

## Data Availability

Not applicable.

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
