# Peer review of "Bacillamide F, Extracted from Marine Bacillus atrophaeus C89, Preliminary Effects on Leukemia Cell Lines"

_biology, 2022, doi:10.3390/biology11121712_

Round 1
Reviewer 1 Report (New Reviewer)
The manuscript of Zhang et al. is well organized and in my opinion, can be published without further modifications.
The manuscript describes the isolation and characterization of a novel non-ribosomal peptide thiazole alkaloid, bacillamide F, isolated from the marine Bacillus atrophaeus C89. This new compound is able to inhibits the proliferation of leukemia cell lines HL60 13 and Jurkat in a dose-dependent manner.
Minor points:
1. Please add description of abbreviations: es. EtOAc, MeOH, etc.
2. In the introduction paragraph the authors need to describe the mechanism of action of the non-ribosomal peptide thiazole alkaloid on leukemia and why bacillamide F differs from the others analogs.
3. I suggest to revise data related to anti-proliferation assay of Jurkat cells. The data relative to 25µM aren’t in line with the others.
Author Response
Comment 1:
The manuscript of Zhang et al. is well organized and in my opinion, can be published without further modifications.
The manuscript describes the isolation and characterization of a novel non-ribosomal peptide thiazole alkaloid, bacillamide F, isolated from the marine Bacillus atrophaeus C89. This new compound is able to inhibits the proliferation of leukemia cell lines HL60 13 and Jurkat in a dose-dependent manner.
Minor points:
- Please add description of abbreviations: es. EtOAc, MeOH, etc.
Reply: We thank the reviewer for the suggestion. We have added the description of abbreviations, on line 79 for Ethyl acetate (EtOAc), on line 81 for Methanol (MeOH), on Line 99 for Nuclear Magnetic Resonance (NMR), on Line 104 for Heteronuclear Multiple Quantum Coherence (HMQC), and on Line 105 for Heteronuclear Multiple Bond Correlation (HMBC), respectively.
- In the introduction paragraph the authors need to describe the mechanism of action of the non-ribosomal peptide thiazole alkaloid on leukemia and why bacillamide F differs from the others analogs.
Reply: We thank the reviewer for the suggestion. In the introduction paragraph, the sentence “Moreover, thiazole derivatives exhibit good anticancer activity with influence on cell cycle, DNA fragmentation and mitochondrial depolarization” has been added on Line 49-51. In the discussion paragraph, the sentence “Bacillamide F has a pyrrole structure (Figure 1b). Pyrrole is a simple heterocyclic and important building block for biologically active compounds [28].” has been added on Line 236 -237.
- I suggest to revise data related to anti-proliferation assay of Jurkat cells. The data relative to 25 µM aren’t in line with the others.
Reply: We thank the reviewer for this valuable insight. We have double checked the original data of bacillamide F in inhibiting the proliferation of Jurkat cells. It shows that we made the anti-proliferation assays of Jurkat and HL60 cells at the same experiment. So it seems that the inhibition of bacillamide F on Jurkat cells at 25 mM is indeed less potent than that of HL60, an effect that we cannot figure out by now. To emphasize this observation, we have now discussed it in the text. The sentence “Intriguingly, Bacillamide F showed a much less inhibitory effect on the proliferation of Jurkat than HL60, e.g. at the concentration of 25 mM (Figure 4), an effect that we can-not explain by now.” has been added on Line 204-206.
Reviewer 2 Report (New Reviewer)
Authors elucidated the structure and cytotoxicity of bacillamide F.
I think those results are worth publishing in Biology.
1. Figure 1a. The structures of bacillamide B and C are wrong. Bacillamide B has a hydroxyl group at C15 and bacillamide C has an acetylamino group. Please change the double bond at C-15.
2. I cannot find Table S1. Did authors submit the Supplementary Materials?
3. The position of the acetyl group was not proved by NMR experiments. HMBC correlations from H15 or H18 to C19 and COSY correlation H10 and NH-2 were not described. There is a possibility that the acetyl group binds to N-2.
Author Response
Comment 2 Authors elucidated the structure and cytotoxicity of bacillamide F.
I think those results are worth publishing in Biology.
- Figure 1a. The structures of bacillamide B and C are wrong. Bacillamide B has a hydroxyl group at C15 and bacillamide C has an acetylamino group. Please change the double bond at C-15.
Reply: We thank the reviewer for suggestion. In the revised manuscript, we have changed the double bond to single bond at C-15 in Figure 1a.
- I cannot find Table S1. Did authors submit the Supplementary Materials?
Reply: We thank the reviewer for suggestion. We have submitted Table S1 in Supplementary materials.
- The position of the acetyl group was not proved by NMR experiments. HMBC correlations from H15 or H18 to C19 and COSY correlation H10 and NH-2 were not described. There is a possibility that the acetyl group binds to N-2.
Reply: We thank the reviewer for suggestion. The definite correlation from H-15 to C-19 of HMBC explains the acetyl group binds to the N-atom of pyrrolidine as illustrated by the modified Figure 2. The sentence “The other significant HMBC cross-peaks from H-13 to C-11, C-12, and C-14, from H-18 to C-15” has been changed to the sentence “The other significant HMBC cross-peaks from H-13 to C-11, C-12, and C-14, from H-18 to C-14 and C-15” on Line 149. The sentence “further verified by HMBC correlation from H-15 to C-19” has been added on Line 152-153.

This manuscript is a resubmission of an earlier submission. The following is a list of the peer review reports and author responses from that submission.
Round 1
Reviewer 1 Report
- While the article is of interest, suggest to better describe the methodology and number of repeat cultures of the cell lines, with and without Bacillamide F. The graphs made for the effects of Bacillamide F analogue on the cell lines are unconventional, with higher doses preceding lower doses and needs to be altered. Evaluation of genetic and transcription factors as well as effects on cell cycle with and without addition of Bacillamide F analogue would improve validity of the data presented. The statements regarding acute lymphoblastic leukemia compared to the treatment of other leukemias (line 71), “acute lymphoblastic leukemia is more problematic to cure” should change to indicate the age, since in children nearly 90% of patients survive the disease. Regarding the manuscript, the English language can be improved. At times, abbreviations are not described prior to their use and their citation precedes the description of the full verbiage (Example, MPLC, line 106). Addition of some of data currently placed in the supplementary materials to the manuscript can improve the article.
Reviewer 2 Report
The manuscript “Selective inhibition of leukemia cell proliferation by bacillamide F produced by marine Bacillus atrophaeus C89” submitted by Shengnan Zhang et al. to the journal Biology presents a study on a new compound from marine Bacillus atrophaeus C89 including: 1) biosynthesis and extraction; 2) structural identification; 3) plausible biosynthesis pathway; 4) toxicity on leukemia cells. The study is important from the point of view of approaches to obtaining new compounds (the use of additives - amino acids). The authors claim novel properties of bacillamides to inhibit cell proliferation in leukemic cells. I hope that the authors are going to determine the mechanisms of inhibition of proliferation of leukemic cells by bacillamide F.
I would like to recommend authors to specify the leukemic cells (Jurkat and HL60) in the manuscript title. If to follow the logic of the title of the manuscript, in the introduction the authors need to start with a description of leukemia disease and cells.
I don't understand why the authors use this reference (20. Gosangi, B.; Davids, M.; Somarouthu, B.; Alessandrino, F.; Giardino, A.; Ramaiya, N.; Krajewski, K. Review of targeted therapy in chronic lymphocytic leukemia: what a radiologist needs to know about CT interpretation. Cancer Imaging. 2018, 18, 13. doi: 337 10.1186/s40644-018-0146-8).
Please, check the captions for Figures (a redundant text). Figure 1, move the structures according to the chronology of their isolation. Figure 3. Line 174, put the ref. 35 after “for bacillamide production”. Line 186, I didn’t find “Scheme 1”. Check, please, in Figure 2.
The authors need to check the correctness of the citations.
I found that ref. 42 is duplicated with ref. 33 (Lines 245 and 250).
Lines 170 and 227. Please, check ref. 32. I did not find confirmation of the statements there.
Round 2
Reviewer 1 Report
The manuscript has been significantly altered and improved. The title of the article does not entirely correspond to the contents of the manuscript. Most of the article is devoted to separation and identification of bacillamide F and only a fraction of the described experiments is devoted to the selective inhibition of a few leukemia cell lines. While the quality of the English language has significantly improved, still there is room for further improvement.
Reviewer 3 Report
Comments on the responses:
1. The introduction needs further revision. The bacillamide discovery paragraph is too wordy and it is not informative at all.
3. Authors should provide the results here in this manuscript, otherwise, the results in this manuscript is not strong enough for publication. Also, please make sure all the assays are supported statistically. Figure 3 is not acceptable.
4. I do not think the biosynthesis in this manuscript is reasonable. Please double check.
Round 3
Reviewer 3 Report
As the reviewer said, it has not displayed completely results, but summarize the research framework of the manuscript.---I think authors should rerun the experiments in Figure 3. At least make sure they provide publishable results.
compound bacillamide F is a novel compound, according to its chemical structure, we have predicted biosynthesis of bacillamide F.-----apparently the biosynthetic pathway proposed by the authors is not reasonable. Bacillamide F is not even a novel compound. I propose two NRPS modules first took Pro and then Cys, cyclization, then tryptamine as a nucleophile to cleavage the thioester bond for the release of the peptide. I do not think this paper is publishable without figuring out the biosynthetic pathway.